# Investigating the application value of the "Forward-Deployed Position" model in operating room support by the central sterile supply department

Qian Yang[1]ᵒ, Yun Wang[1]ᵒ, Mingjun Wu[2]*

1 Central Sterile Supply Department, 940th Hospital of Joint Logistic Support Force of Chinese People's Liberation Army, Lanzhou, Gansu, China, 2 Department of Respiratory and Critical Care Medicine, 940th Hospital of Joint Logistic Support Force of Chinese People's Liberation Army, Lanzhou, Gansu, China

ᵒ These authors contributed equally to this work and share first authorship.
* 515290614@qq.com

## Abstract

### Objective

To investigate the application effects of the "Forward-Deployed Position" model in operating room support provided by the Central Sterile Supply Department (CSSD).

### Methods

This study employed a historical control design. Data from the period when the "Forward-Deployed Position" model was implemented in our hospital in 2024 were set as the observation group, while data from 2023 under the traditional model served as the control group. The two groups were compared regarding indicators including the instrument re-cleaning rate, packaging qualification rate, clinical department satisfaction, and on-time start rate for the first scheduled surgery of the day.

### Results

Compared with the control group, the observation group demonstrated a significant increase in the packaging qualification rate (99.05% vs 98.05%), clinical department satisfaction, and the on-time start rate for the first scheduled surgery (95.05% vs 85.63%), alongside a significant decrease in the instrument re-cleaning rate (0.37% vs 1.05%). All these differences were statistically significant ($p < 0.05$).

### Conclusion

Provided that implementation conditions such as human resources are fully considered, the "Forward-Deployed Position" model can effectively enhance the work quality of the CSSD and the efficiency of surgical support. Its successful implementation

**Data availability statement:** All data underlying the findings reported in this paper are provided within the paper and its Supporting information files. Specifically, the raw data for instrument re-cleaning rates (Table 1) are available in S1 Data; the raw data for instrument packaging qualification rates (Table 2) are available in S2 Data; the raw data for clinical department satisfaction scores (Table 3) are available in S3 Data; and the raw data for on-time start rates for the first scheduled surgery (Table 4) are available in S4 Data. No additional external data repository was used.

**Funding:** The author(s) received no specific funding for this work.

**Competing interests:** The authors have declared that no competing interests exist.

relies on a specialized team and sustained inter-departmental collaboration, offering a valuable reference for optimizing surgical workflow management.

## 1. Introduction

Healthcare-associated infections (HAIs) represent a significant global public health challenge, whose occurrence is closely linked to the quality of medical instrument disinfection and sterilization. According to the World Health Organization, approximately 7 out of every 100 hospitalized patients in high-income countries acquire a HAI, compared with 15 in low- and middle-income countries. Meanwhile, studies have reported HAI prevalence rates in India ranging from 8% to 58% [1]. Inadequate processing of surgical instruments by the Central Sterile Supply Department (CSSD) is a critical factor contributing to HAIs [2]. HAIs not only severely compromise patient safety but also impose a substantial economic burden on both patients and healthcare institutions, while concurrently diminishing the public perception of hospitals [3].

As the core department responsible for the cleaning, disinfection, packaging, sterilization, and supply of medical devices within a hospital [4,5], the CSSD plays an indispensable role in safeguarding patient safety and enhancing surgical efficiency. With continuous advancements in medical technology, surgical instruments are becoming increasingly complex and sophisticated. The processing difficulty of reusable delicate instruments has significantly increased, presenting new challenges for the CSSD [3,6]. Previous research has demonstrated that strengthening quality management in the CSSD is an effective strategy for controlling and reducing HAIs [5].

Under the traditional workflow model, communication between the CSSD and the operating room is often passive, with frequent delays in problem feedback and resolution, which compromises work quality and poses potential medical risks. "Forward Deployment," a reform concept originating from the field of economic management, emphasizes shifting the focus of service and management to the frontline to enhance efficiency and responsiveness. Introducing this concept, the CSSD implemented the "Forward-Deployed Position for Surgical Support," transitioning part of its work from logistical support to the clinical frontline, thereby establishing direct coordination with surgical departments [7]. This model aims to comprehensively improve the quality of surgical support and ensure patient safety by proactively engaging with clinical practice, understanding surgeons' needs, and responding swiftly and effectively to resolve issues. This study aims to systematically investigate the practical application effects of the "Forward-Deployed Position" model in CSSD surgical support, providing evidence for optimizing CSSD functions and enhancing overall healthcare quality and patient safety.

## 2. Materials and methods

### 2.1 General information

This study employed a historical control design. The focus of this study was the support services provided by our hospital's CSSD to the operating rooms. Data from

the period when the "Forward-Deployed Position" model for surgical support was implemented in 2024 were designated as the observation group. Data from 2023 under the traditional support model served as the control group. Data from both groups regarding the instrument re-cleaning rate, packaging qualification rate, clinical department satisfaction, and on-time start rate for the first scheduled surgery were collected for comparative analysis. Ethical review was waived for this study as it was a retrospective analysis of anonymized quality control data with no patient involvement.

## 2.2 Methods

**2.2.1 Establishment and operation of the forward-deployment team.** A specialized six-member forward-deployment team was established through a comprehensive selection process based on criteria including work experience, professional title, specialty nurse certification, sense of responsibility, and communication skills. The team was led by the head nurse as the team leader, with two associate senior nurses as deputy leaders, and three area leaders as support supervisors. The team's operational mechanism was as follows:

**Department Liaison:** The three support supervisors remained in the CSSD, responsible for coordinating internal workflows and implementing corrective actions for identified issues.

**Frontline Support:** The two deputy leaders arrived at the operating room before 08:00 daily. They were responsible for inspecting the sterile instruments and auxiliary packs scheduled for use, verifying their packaging and sterilization quality, and subsequently providing in-room support by circulating within various operating suites.

**Issue Management:** Support problems identified during in-room sessions were communicated face-to-face with operating room staff, aiming for immediate resolution. Issues that could not be resolved on the spot were meticulously documented in a "First-Contact Responsibility" log and tracked persistently until closure. The effectiveness of the corrective actions was ultimately evaluated by the individual who initially reported the problem.

**Training and Communication:** Team members documented typical issues using photographs or videos. At the end of each month, these were used for instructional sessions within the department, conducted via PowerPoint presentations and live simulations. Quarterly, the head nurse led the two deputy leaders in face-to-face meetings with directors of relevant clinical departments to report on issue resolution during the past quarter and collaboratively discuss pending matters. Furthermore, clinical surgeons were regularly invited to deliver lectures on the clinical use and maintenance of specific instruments, enhancing the professional knowledge and practical skills of CSSD personnel.

**2.2.2 Statistical indicators.** This study primarily compared the following four indicators:

Instrument re-cleaning rate = (Number of instruments requiring re-cleaning / Total number of instruments cleaned) × 100%;

Packaging qualification rate = (Number of instruments passing packaging inspection / Total number of instruments packaged) × 100%;

Clinical department satisfaction: Satisfaction survey questionnaires were distributed weekly to each department performing the first scheduled surgery for scoring (0 indicating highly dissatisfied, 10 indicating highly satisfied);

On-time start rate for the first scheduled surgery = (Number of first surgeries starting on time / Total number of first scheduled surgeries) × 100%.

## 2.3 Statistical analysis

Data analysis was performed using SPSS software (version 26.0). Continuous data conforming to a normal distribution are presented as mean ± standard deviation and were compared between groups using the t-test. Continuous data not conforming to a normal distribution are presented as median (interquartile range) and were compared between groups using the Mann-Whitney U test. Categorical data are presented as rates (percentages) and were compared between groups using the chi-square ($\chi^2$) test. A p-value < 0.05 was considered statistically significant.

## 3. Results

### 3.1 Comparison of instrument re-cleaning rates

The instrument re-cleaning rate in the observation group was 0.37% (336/91,765), significantly lower than the 1.05% (756/71,850) in the control group. This difference was statistically significant ($\chi^2 = 286.080$, $p < 0.001$). Detailed data are presented in Table 1. The raw data underlying this table are available in S1 Data.

### 3.2 Comparison of instrument packaging qualification rates

The instrument packaging qualification rate in the observation group was 99.05% (90,890/91,765), higher than the 98.05% (70,452/71,850) in the control group. This difference was statistically significant ($\chi^2 = 289.585$, $p < 0.001$). Detailed data are presented in Table 2. The raw data underlying this table are available in S2 Data.

### 3.3 Comparison of clinical department satisfaction

The clinical department satisfaction scores showed that the median score in the observation group (9.0 points) was higher than that in the control group (8.0 points). The Mann-Whitney U test indicated that the difference between the two groups was statistically significant ($z = -17.623$, $p < 0.001$). Detailed data are presented in Table 3. The raw data underlying this table are available in S3 Data.

### 3.4 Comparison of on-time start rate for the first scheduled surgery

The on-time start rate for the first scheduled surgery in the observation group was 95.05% (6,939/7,300), significantly higher than the 85.63% (6,251/7,300) in the control group. This difference was statistically significant ($\chi^2 = 371.592$, $p < 0.001$). Detailed data are presented in Table 4. The raw data underlying this table are available in S4 Data.

## 4. Discussion

Utilizing a historical control comparison, this study systematically evaluated the application efficacy of the Forward-Deployed Position model within the surgical support framework of the Central Sterile Supply Department. Our findings demonstrate that the implementation of this model yielded significant positive impacts across several key metrics: reducing the instrument recirculation rate, enhancing the packaging qualification rate, improving clinical department satisfaction, and increasing the on-time initiation rate for the first scheduled surgery of the day. These results not only corroborate the

**Table 1. Comparison of instrument re-cleaning between the two groups.**

| Group | Total Instruments Cleaned (n) | Re-cleaned Number (%) | Not Re-cleaned Number (%) | $\chi^2$ | p-value |
|---|---|---|---|---|---|
| Control Group | 71,850 | 756 (1.05) | 71,094 (98.95) | 286.080 | <0.001 |
| Observation Group | 91,765 | 336 (0.37) | 91,429 (99.63) | | |

**Table 2. Comparison of instrument packaging qualification between the two groups.**

| Group | Total Instruments Packaged (n) | Qualified Number (%) | Non-qualified Number (%) | $\chi^2$ | p-value |
|---|---|---|---|---|---|
| Control Group | 71,850 | 70,452 (98.05) | 1,398 (1.95) | 289.585 | <0.001 |
| Observation Group | 91,765 | 90,890 (99.05) | 875 (0.95) | | |

**Table 3. Comparison of clinical satisfaction between the two groups.**

| Group | Sample Size (n) | Satisfaction Score [Median (IQR)] | z-value | p-value |
|---|---|---|---|---|
| Control Group | 240 | 8.0 (8.0, 8.0) | −17.623 | <0.001 |
| Observation Group | 240 | 9.0 (9.0, 9.0) | | |

**Table 4. Comparison of on-time start status for the first surgeries between the two groups.**

| Group | Total First Surgeries (n) | On-time Starts (%) | Delayed Starts (%) | χ² | p-value |
|---|---|---|---|---|---|
| Control Group | 7,300 | 6,251 (85.63) | 1,049 (14.37) | 371.592 | <0.001 |
| Observation Group | 7,300 | 6,939 (95.05) | 361 (4.95) | | |

general value of shifting management focus to the frontline but also illuminate the unique mechanisms through which this model facilitates the restructuring of intra-hospital collaborative workflows and enables a paradigm shift from reactive support to proactive service.

## 4.1 Establishment of a closed-loop quality control system

The most definitive finding of this study—the significant improvement in both instrument recirculation rate and packaging qualification rate—strongly supports and extends existing research on quality management in the CSSD. While previous literature consistently identifies cleaning and packaging as fundamental barriers for successful sterilization [4,6,8], the contribution of this study lies in elucidating how the "Forward-Deployed Position" model reinforces this barrier through a reconfiguration of physical space and professional roles. In the traditional model, where the CSSD and operating rooms are physically separate, quality issues are often only identified during instrument use, resulting in a "delayed feedback" loop. By deploying specialized personnel directly within the surgical environment, our model enables the immediate identification, communication, and correction of issues. This approach not only represents a practical application of the "Key Point Control Theory" but also establishes a real-time, closed-loop quality control cycle [3,9]. On-site support in the operating rooms allows for instruments with inadequate cleaning to be identified prior to packaging, and packaging defects to be rectified before the surgery begins, thereby substantially advancing the interception point for quality defects. Consequently, our results not only confirm the efficacy of refined management practices in enhancing CSSD quality indicators but also clarify the mechanism for its achievement: by breaking down the physical and informational silos between departments, thereby transforming terminal quality control into process quality control [5,10].

## 4.2 Value reconstruction: from "item supply" to "process integration"

Beyond the anticipated improvements in internal quality, a more pivotal finding of this study is the profound impact of the Forward-Deployed Position model on overall surgical efficiency. The nearly 10-percentage-point surge in the on-time start rate for the first scheduled surgery (85.63%→95.05%) signifies an impact extending beyond the CSSD's immediate operational scope, indicating optimization of cross-departmental system efficiency. This outcome, partly exceeding the initial expectation of merely improving instrument quality, suggests the model's value lies not only in "supplying qualified items" but also in ensuring the "smooth operation of the surgical workflow." To explain this, we propose a novel interpretative framework: the Forward-Deployed Position restructures the responsiveness of the surgical supply chain by mitigating information asymmetry and enabling collaborative decision-making. In the traditional model, the CSSD, functioning as a backend support department, experienced a lag in perceiving the dynamic needs of the operating rooms [11]. In contrast, the embedded presence of forward-deployed personnel within the surgical setting enables the direct capture of critical

information, such as discrepancies between instrument sets and actual surgical requirements. Of the 260 issues resolved in 2024, a substantial 75% (n = 195) pertained to instrument adjustments, demonstrating the tangible value of this on-site intelligence. Through optimized adjustments to instrument sets (e.g., streamlining 2–3 sets down to 1), the outcomes achieved not only conserved manpower and material resources but, more profoundly, reduced the complexity of surgical preparation and accelerated instrument turnover. This directly deconstructs systemic bottlenecks contributing to surgical delays. This indicates that the efficacy of the Forward-Deployed Position has evolved beyond the realm of "quality control" to encompass "process integration" and "system optimization," transforming the CSSD from a passive item provider into an active enabler of the surgical workflow.

### 4.3 Interplay between professional value recognition and systemic transformation

The marked improvement in clinical satisfaction, coupled with the enhanced sense of value among nurses, collectively substantiates the positive effects of this model at the organizational behavior level. This outcome is not attributable to a single factor but rather results from the synergistic interplay of the multiple mechanisms described above. The increase in satisfaction stems from reliable product quality, efficient problem resolution, and streamlined surgical workflows, which collectively foster constructive inter-departmental trust. Meanwhile, CSSD nurses, through their deep immersion in clinical settings and direct involvement in problem-solving, find their specialized knowledge applied and recognized on a broader stage. This fosters heightened professional identity and a greater sense of value [12]. This enhancement in perceived value is inextricably linked to the role transformation of CSSD work from "behind-the-scenes" to "center-stage," establishing a positive feedback cycle of "efficacy enhancement→value recognition→motivation reinforcement."

### 4.4 Limitations and future directions

This study has several limitations. Firstly, as a single-center study, the generalizability of its findings requires cautious consideration. Secondly, the model's success is highly dependent on the comprehensive competencies of the forward-deployed staff; these specific human resource requirements must be fully accounted for during broader implementation. Future research should investigate the minimum resource thresholds for implementing this model across different tiers of hospitals and employ qualitative methods to deeply analyze the key organizational and cultural factors influencing the effectiveness of inter-departmental collaboration. Additionally, as this study employed a historical control design (comparing data from 2024 with those from 2023), the observed improvements may be partially attributable to unmeasured confounding factors such as annual variations in surgical caseload, fluctuations in staffing, or institutional policy changes during the study period. Although the implementation of the Forward-Deployed Position model was the primary intervention, we acknowledge that the absence of concurrent controls limits our ability to definitively attribute the observed effects solely to the intervention. Future studies employing interrupted time series analysis or prospective controlled designs would help strengthen causal inference by accounting for temporal trends and potential confounders.

### 5. Conclusion

In summary, the effectiveness of the Forward-Deployed Position model demonstrates a progressive evolution from quantitative improvements to a qualitative leap in efficacy. Initially, by establishing a real-time quality control loop, it verifies and achieves optimization of fundamental quality indicators in the CSSD. Subsequently, and unexpectedly, it accomplishes a value transformation from material supply assurance to surgical workflow integration through the reduction of information barriers and optimization of resource allocation. The success of this model provides new insights for hospital management: enhancing operating room efficiency depends not only on advances in surgical techniques but equally on support systems like the CSSD unleashing their latent systemic value through proactive role transformation and process reengineering. Its successful implementation relies on specialized teams and sustained interdepartmental collaboration, making this model a valuable reference for optimizing surgical workflow management.

## Supporting information

**S1 Data. Raw data of instrument re-cleaning rates.** This table presents the raw data underlying Table 1, showing the total number of cleanings, the number of re-cleanings, and the calculated re-cleaning rates for the control group (2023) and the observation group (2024). The control group had 71,850 cleanings with 756 re-cleanings (1.05%), while the observation group had 91,765 cleanings with 336 re-cleanings (0.37%).
(DOCX)

**S2 Data. Raw data of instrument packaging qualification rates.** This table presents the raw data underlying Table 2, showing the total number of packages, the number passed, and the calculated packaging qualification rates for the control group (2023) and the observation group (2024).
(DOCX)

**S3 Data. Raw data of clinical department satisfaction scores.** This table presents the raw data underlying Table 3, showing monthly satisfaction scores (0–10 scale) for the control group (2023) and the observation group (2024).
(XLSX)

**S4 Data. Raw data of on-time start rate for the first scheduled surgery.** This table presents the raw data underlying Table 4, showing the total number of first scheduled surgeries, the number of on-time starts, and the number of delayed starts for the control group (2023) and the observation group (2024).
(DOCX)

## Acknowledgments

We appreciate the anonymous reviewers for their constructive comments that helped improve this manuscript.

## Author contributions

**Conceptualization:** Mingjun Wu, Qian Yang.

**Data curation:** Yun Wang.

**Methodology:** Yun Wang.

**Writing – original draft:** Qian Yang.

**Writing – review & editing:** Mingjun Wu.

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
