## [Decision Letter · Decision Letter 0]

17 Feb 2026

PONE-D-25-62281Investigating the Application Value of the "Forward-Deployed Position" Model in Operating Room Support by the Central Sterile Supply DepartmentPLOS One

Dear Dr. Wu,

Thank you for submitting your manuscript to PLOS ONE. After careful consideration, we feel that it has merit but does not fully meet PLOS ONE’s publication criteria as it currently stands. Therefore, we invite you to submit a revised version of the manuscript that addresses the points raised during the review process.

**ACADEMIC EDITOR: Please address comments and concerns araised by reviewer 1. In addition, we may invite a new reviewer to join reviewer 1 in the next round of review.**

We look forward to receiving your revised manuscript.

Kind regards,

Ziyu Qi, M.D., Ph.D.

Academic Editor

PLOS One

Journal Requirements:

4. In the online submission form, you indicated that data cannot be shared publicly because of [restrictions imposed by the institutional ethics committee to protect patient confidentiality]. Data are available from the 940th Hospital of Joint Logistic Support Force of Chinese People's Liberation Army Institutional Data Access / Ethics Committee (contact via the corresponding author at 515290614@qq.com) for researchers who meet the criteria for access to confidential data.

6. We note that the grant information you provided in the ‘Funding Information’ and ‘Financial Disclosure’ sections do not match.

7. Please update your submission to use the PLOS LaTeX template. The template and more information on our requirements for LaTeX submissions can be found at http://journals.plos.org/plosone/s/latex.

Reviewers' comments:

Reviewer's Responses to Questions

**Comments to the Author**

1. Is the manuscript technically sound, and do the data support the conclusions?

Reviewer #1: Yes

2. Has the statistical analysis been performed appropriately and rigorously? 

Reviewer #1: Yes

3. Have the authors made all data underlying the findings in their manuscript fully available?

Reviewer #1: Yes

4. Is the manuscript presented in an intelligible fashion and written in standard English?

Reviewer #1: Yes

5. Review Comments to the Author

Reviewer #1: It is recommended that the manuscript be re-edited and the references updated.It is recommended that the manuscript be re-edited and the references updated.It is recommended that the manuscript be re-edited and the references updated.

6. PLOS authors have the option to publish the peer review history of their article (what does this mean?). If published, this will include your full peer review and any attached files.

Reviewer #1: No

---

## [Author Response · Author response to Decision Letter 1]

26 Mar 2026

Dear Editor and Reviewers,

Thank you for your valuable comments and suggestions regarding our manuscript (ID: PONE-D-25-62281). We have carefully considered each point and have revised the manuscript accordingly. Below, we provide a point-by-point response to the comments.

1. PLOS ONE style requirements

We have reviewed the PLOS ONE style templates and confirmed that our manuscript meets the journal’s formatting requirements, including file naming conventions.

2. Ethics statement

We have moved the ethics statement to the Methods section, as requested. The statement no longer appears in any other section of the manuscript.

3–5. Data availability and sharing restrictions

We appreciate the journal’s emphasis on data transparency. In response to the comments regarding data sharing, we have now uploaded the de-identified dataset as Supporting Information files. Accordingly, we have updated the Data Availability Statement in the submission form to reflect that all underlying data are provided within the Supporting Information files.

We confirm that no ethical or legal restrictions prevent the sharing of this de-identified dataset, and therefore no exemption request is needed.

6. Funding information

We have corrected the inconsistency between the “Funding Information” and “Financial Disclosure” sections.

7. LaTeX template

We have updated our submission to use the PLOS LaTeX template, in accordance with the journal’s requirements.

8.It is recommended that the manuscript be re-edited and the references updated.

We have carefully revised the manuscript and updated all references to ensure they are current. The references now predominantly include articles published within the last five years. Additionally, we have standardized the citation format to comply with PLOS ONE style requirements, including the addition of missing DOIs, PMIDs, and PMCID numbers where applicable. We believe these revisions have improved the quality and timeliness of the literature cited.

We believe that these revisions have addressed all the concerns raised. We appreciate your time and effort in handling our manuscript and look forward to your further feedback.

Sincerely,

Mingjun Wu

515290614@qq.com

---

## [Editor Report · Decision Letter 1]

20 Apr 2026

Investigating the application value of the "Forward-deployed position" model in operating room support by the central sterile supply department

PONE-D-25-62281R1

Dear Dr. Wu,

We’re pleased to inform you that your manuscript has been judged scientifically suitable for publication and will be formally accepted for publication once it meets all outstanding technical requirements.

Kind regards,

Ziyu Qi, M.D., Ph.D.

Academic Editor

PLOS One
---

## [Editor Report · Acceptance letter]

PONE-D-25-62281R1

PLOS One

Dear Dr. Wu,

I'm pleased to inform you that your manuscript has been deemed suitable for publication in PLOS One. Congratulations! Your manuscript is now being handed over to our production team.

Kind regards,

on behalf of

Dr. Ziyu Qi

Academic Editor

PLOS One